# Pre-Sowing Irrigation Plus Surface Fertilization Improves Morpho-Physiological Traits and Sustaining Water-Nitrogen Productivity of Cotton

**Zongkui Chen [†], Hongyun Gao [†], Fei Hou, Aziz Khan *** and **Honghai Luo ***

Key Laboratory of Oasis Eco-Agriculture, Xinjiang Production and Construction Group, Shihezi University, Shihezi 832003, China; chenzongkui90@foxmail.com (Z.C.); gaohongyun1995@163.com (H.G.); houfeishzu@163.com (F.H.)
* Correspondence: azizkhanturlandi@gmail.com (A.K.); luohonghai79@163.com (H.L.); Tel.: +156-7670-1561 (A.K.); +133-4546-6760 (H.L.)
† The authors contributed equally to this article.

**Abstract:** The changing climatic conditions are causing erratic rains and frequent episodes of moisture stress; these impose a great challenge to cotton productivity by negatively affecting plant physiological, biochemical and molecular processes. This situation requires an efficient management of water-nutrient to achieve optimal crop production. Wise use of water-nutrient in cotton production and improved water use-efficiency may help to produce more crop per drop. We hypothesized that the application of nitrogen into deep soil layers can improve water-nitrogen productivity by promoting root growth and functional attributes of cotton crop. To test this hypothesis, a two-year pot experiment under field conditions was conducted to explore the effects of two irrigation levels (i.e., pre-sowing irrigation ($W_{80}$) and no pre-sowing irrigation ($W_0$)) combined with different fertilization methods (i.e., surface application ($F_{10}$) and deep application ($F_{30}$)) on soil water content, soil available nitrogen, roots morpho-physiological attributes, dry mass and water-nitrogen productivity of cotton. $W_{80}$ treatment increased root length by 3.1%–17.5% in the 0–40 cm soil layer compared with $W_0$. $W_{80}$ had 11.3%–52.9% higher root nitrate reductase activity in the 10–30 cm soil layer and 18.8%–67.9% in the 60–80 cm soil layer compared with $W_0$. The $W_{80}F_{10}$ resulted in 4.3%–44.1% greater root nitrate reductase activity compared with other treatments in the 0–30 cm soil layer at 54–84 days after emergence. Water-nitrogen productivity was positively associated with dry mass, water consumption, root length and root nitrate reductase activity. Our data highlighted that pre-sowing irrigation coupled with basal surface fertilization is a promising option in terms of improved cotton root growth. Functioning in the surface soil profile led to a higher reproductive organ biomass production and water-nitrogen productivity.

**Keywords:** cotton; dry matter yield; root growth; root physiology; water productivity; nitrogen productivity

## 1. Introduction

Cotton is a commercial cash crop providing fiber, oil, and animal feed globally [1]. With the increasing population comes an increased demand for food and fiber, but the threats of climate change are challenging crop production. Crop intensification to produce more food, fiber and feed needs more water, but water resources are limited. Although cotton is considered a drought resistance crop, its productivity is negatively affected by drought stress and nutrient deficiency which results in reduced growth, physiological, biochemical and molecular events [2,3]. Drought stress causes a 50% to

73% reduction in cotton yield [4]. Transgenic cotton cultivars are more susceptible to moisture deficit conditions [3]. Therefore, lower water availability has threatened the productivity of irrigated cotton ecosystem. Hence, strategy to increase water conservation and nutrient uptake are needed to achieve optimal cotton yield [5,6].

Water-nitrogen productivity and cotton production can be improved by application of water-nutrient at the proper growth period of cotton crop [7,8]. However, many water-nutrient conservations strategies can lead to unbalanced organs development such as, the competition between root and aerial plant part (mainly reproductive organs), thus vegetative organs growth surpass reproductive organs development, which in turn decreased water productivity and yield. Moreover, greater above ground dry matter accumulation, especially in reproductive organs can drive cotton yield [9]. An excessive root expansion can reduce growth of aerial plant parts [10,11], but lower root dry matter accumulation affects root distribution and physiological activity in the soil [12,13]. Therefore, it is essential to enhance cotton root activity and distribution in the soil to achieve higher water-nutrient productivity via balancing the growth and development between aerial and underground parts of cotton plant.

Root morphology and physiology are closely associated with the growth and development of aboveground plants. The rates and modes of water and nutrient application influences crop growth and water-nutrient productivity [11,14,15] by affecting root morphological and physiological activity [16]. Poor irrigation practices can develop a large root system and induce aging signals (such as, ABA) that can lead to low dry matter accumulation and water-nutrient productivity [17–19]. An efficient water-nitrogen management can enhance root functioning, increases water-nitrogen absorption, which in turn promote reproductive organ dry matter accumulation and water-nutrient productivity [16,20]. Hence, facilitating the relationship between root and water-nutrient in the root zone is essential for improving water-nutrient productive potential of reproductive organs to achieve higher water-nutrient use efficiency.

Xinjiang is the major cotton growing province in China, contributing 67% to the total national lint production [21], where low water availability and poor nutrient management have imposed a great challenge to cotton production. In cotton, root development occurs before full flowering stage and is mainly affected by soil moisture and basal fertilization. Post-sowing irrigation and snow melt can enrich deep water layer (important soil moisture storage) in the soil. This can lead to a deeper root growth, enhance water uptake, improve photosynthetic capacity and reduces irrigation frequency [20,22]. Basal fertilization can promote root growth and increase nutrient availability [23,24]. Single effects of deep water layer [20] and basal fertilization [23] on cotton root have been documented, but the effect of combine application on cotton root growth and physiology in different soil profile to regulate water-nitrogen productivity is elusive. The aim of this study was firstly to determine the effects of pre-sowing irrigation and basal fertilization on soil water content, available nitrogen, root morpho-physiological traits and above dry mass production and secondly to analyze the relationship between root growth and water-nutrient productivity in the root zone of cotton crop.

## 2. Materials and Methods

### 2.1. Details of Experimental Site

A two-year pot experiment under field conditions was conducted at the research station of Shihezi University Xinjiang, China (45°19′ N, 74°56′ E) during 2015 and 2016 growing seasons. In the region, evapotranspiration was 1425 mm. The mean rainfall and temperatures in both years are presented in Figure 1. Cotton cultivar Xinluzao 45 seeds were sown in polyvinyl chloride (PVC) tubes (diameter, 30 cm; the tubes consisted of three stacked sections; each section was 40 cm high with 120 cm height). The bottom of the tube was covered with a wire to hold soil. The soil was clay loam comprised of 1.43 g m$^{-3}$ bulk density, 24.6% field capacity, 7.6 pH, 54.9 mg kg$^{-1}$ alkali hydrolysable N, 16.8 mg kg$^{-1}$ Olsen-P, 196 mg kg$^{-1}$ exchangeable K and 12.5 g kg$^{-1}$ organic matter.

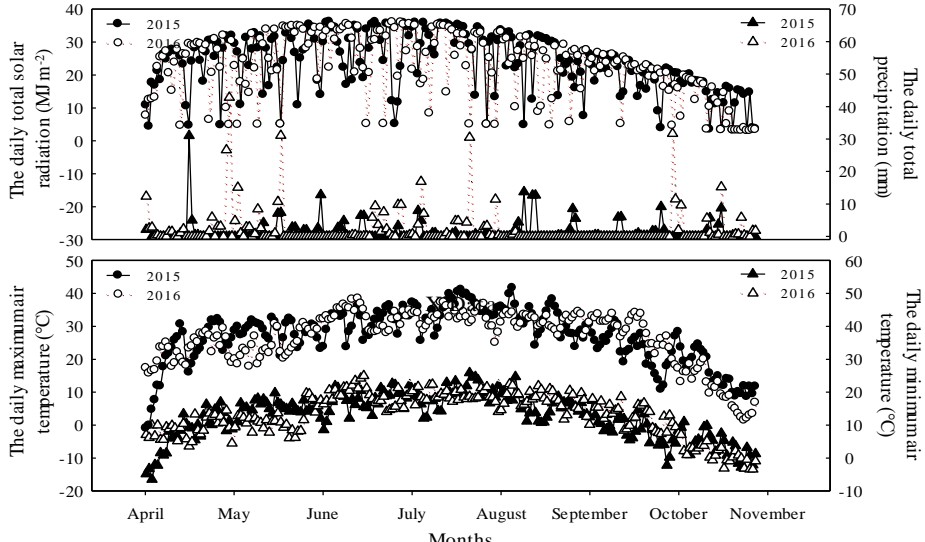

**Figure 1.** The daily total solar radiation (MJ m$^{-2}$), total precipitation (mm), maximum and minimum air temperature (°C) during cotton growing season in Shihezi (2015–2016).

### 2.2. Experimental Design and Crop Management

A randomized complete block design was employed with four treatments with 32 tubes per treatment. Irrigation treatments were: pre-plant irrigation (W$_{80}$, watered with 0.28 m$^3$ (80 ± 5% of field capacity) per tube before sowing), no pre-plant irrigation (W$_0$, no water was applied over the entire depth of the tube) with two fertilization depth (i.e., surface application (F$_{10}$, sufficient basal fertilizer in the 10–20 cm soil layer before sowing and deep fertilization (F$_{30}$, sufficient basal fertilizer in the 30–40 cm layer before sowing)) in each tube. Nitrogen (N) was applied at the ratio of 1:4 as basal fertilizer by topdressing method. Phosphorus (P$_2$O$_5$) and potassium (K$_2$O) were supplemented as basal fertilization. Urea (CO(NH$_2$)$_2$, 46.0% N) at the rate of 13.8 g per tube was used for N and mono-potassium phosphate at the rate of 18 g ((KH$_2$PO$_4$) 52.0% P$_2$O$_5$ and 35.4% K$_2$O) was used per tube as P$_2$O$_5$ and K$_2$O.

Four seeds per tube were sown at a depth of 3 cm on April 25th and May 1st in 2015 and 2016 growing season. Seeds were placed 10 cm apart in one direction and 20 cm apart in another direction. Four seedlings were left per tube. Drip laterals (Beijing Lvyuan Inc., Beijing, China) were installed on the top of each tube with a single emitter. The top of the tube was covered with a polyethylene film to reduce evaporation. Each pot was drip-irrigated each after four days. The total amount of water supplied to the plants was 434 mm each year. Standard local pest control measures were adopted in both cropping seasons.

### 2.3. Observations

During both years, soil water content, available N, dry matter accumulation, root morphological and physiological traits were assessed at 39, 54, 69, 84 and 99 days after emergence (DAE).

### 2.4. Soil Water Content and Available Nitrogen

The irrigation amount during growth period was based on measurement of the soil moisture content in the 0–40 cm soil layer using the Time Domain Reflectometry (TDR). Water supplied to the crop can be defined as:

$$A = (W_p - W_a) \times H \tag{1}$$

where *A* is the volume of water supplied (mm) and $W_p$ is the field capacity in the 0–40 cm soil profile. $W_a$ is the average relative soil moisture content in the 0–40 cm soil profile that was measured by TDR and H is the thickness of the soil layers using drip irrigation system (mm). Changes in soil moisture

content in the 0–120 cm soil profile was determined by the stoving method. During root sampling, fresh soil samples were immediately collected from each soil layer (i.e., 20 cm or 10 cm) in each tube of three replications in 2015 and 2016 growing seasons. Soil was weighted then dried at 85 °C for constant weight. Soil moisture content was expressed as moisture content (g) per dry soil (g). Soil available N was determined by the alkaline hydrolysis diffusion method [25] and was expressed in mg kg$^{-1}$ dry soil.

### 2.5. Root Growth Traits

Three tubes (each treatment) were carefully dug out from the ground level and cut down into 20 cm segments in 2015 and 10 cm segments in 2016 growing season. The segments were immersed in the water for 1 h; roots were placed on a 0.5 mm sieve and rinsed with running water. Plant debris such as weeds and dead roots were separated from 'living' roots according to Gwenzi et al. [26]. The live roots were placed in denoised water and stored in a refrigerator at 4°C for further analysis. Live roots were evenly spread on a plastic tray with deionized water and scanned using a flatbed scanner (300 dpi). Root images were obtained using WinRhizo image analysis software (Regent Instruments, Quebec, Canada). The software was configured to measure root length and then roots were oven-dried at 85 °C for 48 h and weighed.

### 2.6. Root Nitrate Reductase Activity (NR)

Nitrate reductase activity was measured according to Zhou et al. [27] method. Roots were homogenized in extraction buffer and centrifuged for 15 min at 4000 rpm. The supernatants were collected and added to the reaction buffer. After incubation at 25 °C for 30 mins, the reaction was suspended by 1 mL 1% sulphanilamide. The mixture was further centrifuged for 5 mins at 5000 rpm and N-(1-naphthyl) ethylenediamine dihydrochloride was added; the supernatant was used to assess nitrite production at 540 nm after. Root NR activity was expressed as nitrite production (μg) 1 g fresh root per hour.

### 2.7. Biomass Accumulation

To determine cotton biomass accumulation, three tubes (12 plants), in each treatment were chosen and cut down at the cotyledon node during each sampling day. Plant samples were dissected into leaves, stems, buds, flowers, bolls and roots. These samples were oven-dried at 85 °C for 48 h and weighed to a constant weight. A logistic function was used to describe the progress of biomass accumulation [28,29]:

$$Y = \frac{K}{1 + ae^{bt}} \tag{2}$$

In the formula, $t$ (d) is the number of days after emergence (DAE), Y (g) is the biomass at t, K (g) is the maximum biomass while a and b are the constants.

Based on Formula (2), we could calculate:

$$t_1 = \frac{1}{b}\ln\left(\frac{2 + \sqrt{3}}{a}\right) \tag{3}$$

$$t_2 = \frac{1}{b}\ln\left(\frac{2 - \sqrt{3}}{a}\right) \tag{4}$$

$$t_m = -\frac{lna}{b} \tag{5}$$

$$V_m = -\frac{bK}{4} \tag{6}$$

$$V_t = \frac{Y_2 - Y_1}{t_2 - t_1} \tag{7}$$

where $V_m$ (g d$^{-1}$) is the highest biomass accumulation rate; $t_m$ (d) is the largest biomass accumulation period, beginning at $t_1$ and terminating at $t_2$. The factors $Y_1$ and $Y_2$ represent biomass at $t_1$ and $t_2$; $V_t$ is the average biomass accumulation from $t_1$ to $t_2$.

### 2.8. Water-Nitrogen Productivity

Nitrogen productivity was defined as the total biomass (g plant$^{-1}$) or the biomass of each plant organ (root, stem and leaf, bud and boll) per unit of applied fertilizer-nitrogen (g plant$^{-1}$) at different growth stages [30]. In this study, nitrogen productivity was assessed at 39, 54, 69, 84 and 99 DAE.

Water productivity and soil moisture consumption rates were calculated at 39, 54, 69, 84 or 99 DAE according to the method described by Luo et al. [20]. Water productivity is the total biomass (g plant$^{-1}$) or the biomass of each organ (root, stem and leaf, bud and boll) per unit water consumption (cm$^3$ plant$^{-1}$). Moisture consumption rate was calculated according to Luo et al. [20].

### 2.9. Statistical Analysis

Analysis of variance (ANOVA), path analysis was performed using SPSS software version 16.0 (SPSS Inc., Chicago, IL, USA). Correlation analysis was performed using the "heatmap" package in R version 3.5.2. Treatments were separated using the least significant difference (*LSD*) tests at $p \leq 0.05$. Figures were constructed using Sigma Plot software version 10.0 (Systat Software Inc., San Jose, CA, USA). Data represent means ± SD.

## 3. Results

### 3.1. Soil Water Content and Available Nitrogen

Soil moisture content increased by 30.8%–53.1% for $W_{80}$ treatment compared with $W_0$ in the 40–120 cm soil layer prior to 84 DAE (Figure 2). Water consumption of $W_{80}$ was 28.1% more than that of $W_0$ in the 0–40 cm soil profile during whole growth period. No significant differences were observed between $F_{10}$ and $F_{30}$ treatment.

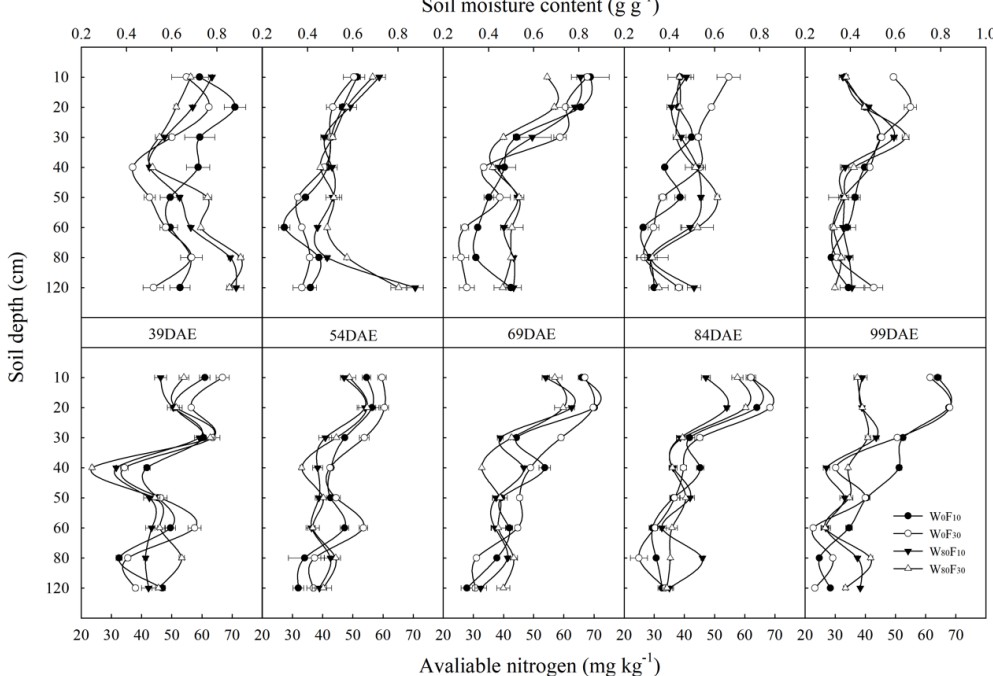

**Figure 2.** Changes in the soil moisture and the available nitrogen in different vertical soil layer (from 0 to 120 cm soil layer) at pre-sowing irrigation ($W_{80}$) or no pre-sowing irrigation ($W_0$) and base fertilizer surface ($F_{10}$) or deep ($F_{30}$) application with the days after emergence in 2016. Bars indicate SD (*n* = 3).

Under $W_{80}$ treatment soil available N decreased by 22% in the 0–40 cm soil layer throughout the growth period (Figure 2) but increased by 7.6% in the 60–120 cm soil layer compared with $W_0$ treatment. No significant differences were observed in the 40–60 cm soil layer. $F_{10}$ treatment had 0.8% and 13.0% lower soil available N compared with $F_{30}$ treatment in the 0–30 cm and 60–80 cm soil layer before 84 DAE, while other soil layer remained unaffected.

*3.2. Root Length*

Cotton plant root length was significantly affected by irrigation levels and fertilization during the whole growth period. Root length gradually increased with the plant development but decreased later in the season (Figure 3). $W_{80}$ treatment increased root length by 3.1–17.5% in the 0–40 cm soil layer but decreased by 7.7–66.1% in the 40–120 cm soil layer after 54 DAE than $W_0$ treatment. $W_{80}$ $F_{10}$ treatment had 3.5%–29.5% higher root length in the 0–40 cm soil layer, but 1.2%–10.5% lowered root length was observed in the 40–120 cm soil layer after 54 DAE compared with $W_0$ $F_{30}$.

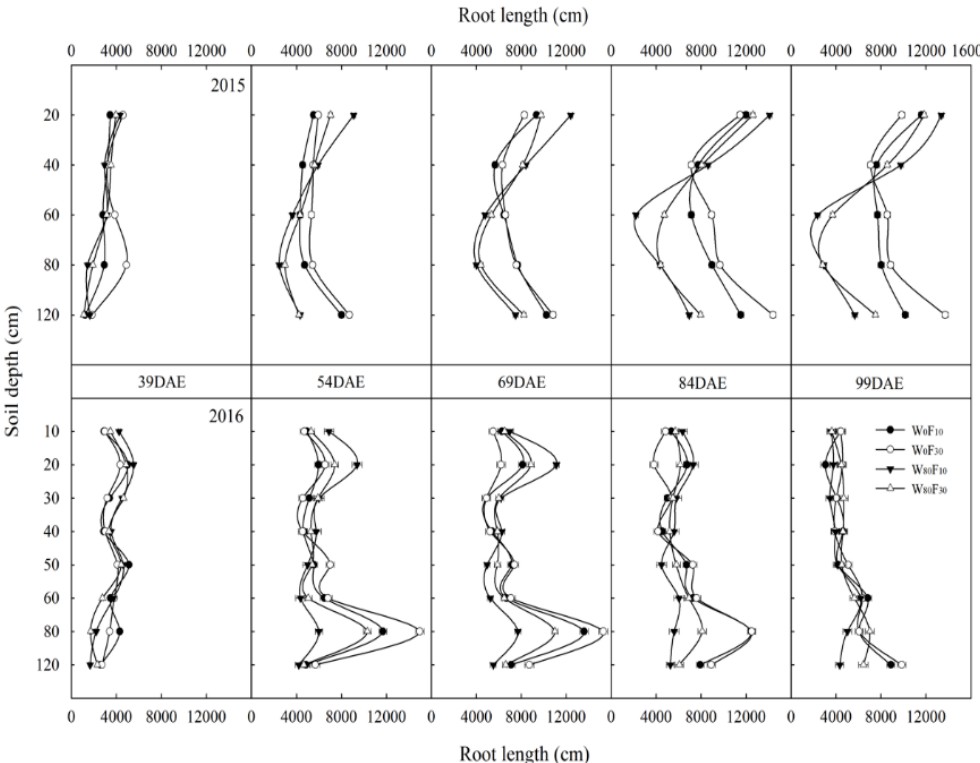

**Figure 3.** Changes in the root length in different vertical soil layer (from 0 to 120 cm soil layer) at pre-sowing irrigation ($W_{80}$) or no pre-sowing irrigation ($W_0$) and basal surface fertilization ($F_{10}$) or deep ($F_{30}$) application with the days after emergence (DAE).

*3.3. Root Nitrate Reductase Activity*

Nitrate reductase activity was rose with the plant development but gradually decreased in the 0–10 cm layer (Figure 4). Compared with $W_{80}$ treatment, nitrate reductase activity in $W_0$ increased by 11.3%–52.9% and 18.8%–67.9%, respectively in the 0–40 and 60–120 cm soil depth at each growth stage but decreased by 13.5%–24.0% in the 40–60 cm soil profile. $F_{10}$ treatment had 4.3%–44.1% and 7.2–18.3% higher nitrate reductase activity in the 10–20 cm and 40–60 cm soil layer soil profile at 54 to 84 and prior 69 DAE over $F_{30}$ fertilization.

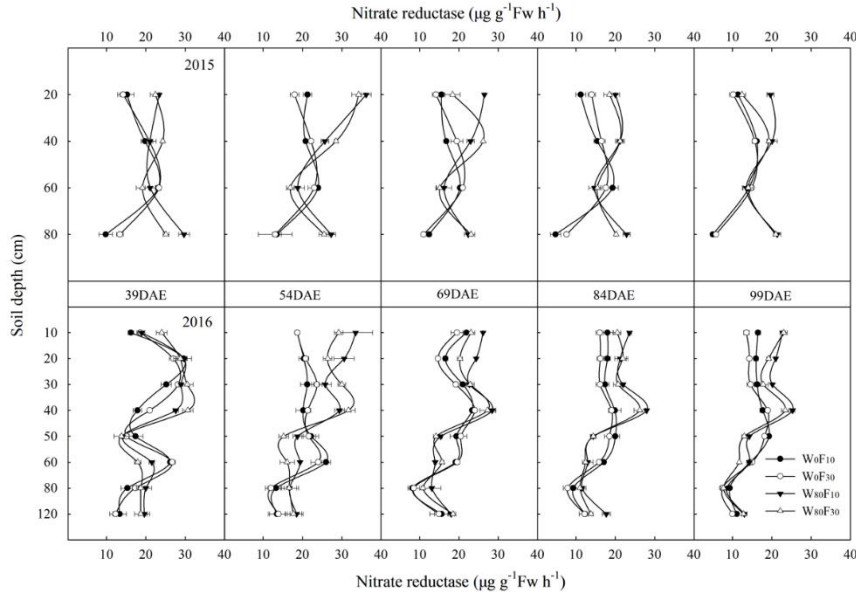

**Figure 4.** Changes in the nitrate reductase activity ($\mu g\ g^{-1}FW\ h^{-1}$) in different vertical soil layer (from 0 to 120 cm soil layer) at pre-sowing irrigation ($W_{80}$) or no pre-sowing irrigation ($W_0$) and basal surface fertilization ($F_{10}$) or deep ($F_{30}$) application with the days after emergence (DAE) in 2015 and 2016. Bars indicate SD ($n = 3$).

### 3.4. Cotton Plant Biomass Accumulation

Nitrogen and irrigation application method significantly altered cotton plant vegetative and reproductive organs biomass accumulation during both years (Table 1). Root, stem plus leaf, bolls and total plant biomass accumulation increased by 11.6%, 30.5%, 48.2% and 22.4%, respectively, in $W_{80}$ over $W_0$ treatment. A 10% and 2% higher root and reproductive organs biomass produced in $W_{80} F_{10}$ treatment compared with $W_{80} F_{30}$ during both growing seasons.

**Table 1.** Changes in vegetative and reproductive and total organ biomass accumulation under different irrigation and fertilization during 2015 and 2016.

| Years | Treatments | Root Dry Matter (g plant$^{-1}$) | Stem and Leaf Dry Matter (g plant$^{-1}$) | Bud and Boll Dry Matter (g plant$^{-1}$) | The Total Dry Matter (g plant$^{-1}$) |
|---|---|---|---|---|---|
| 2015 | $W_0F_{10}$ | 19.8 ± 0.42 c | 19.6 ± 0.05 c | 17.4 ± 0.59 c | 62.06 ± 0.48 b |
| | $W_0F_{30}$ | 19.9 ± 0.29 c | 16.2 ± 0.27 d | 13.1 ± 0.9 d | 53.47 ± 1.53 c |
| | $W_{80}F_{10}$ | 23.2 ± 0.19 a | 22.1 ± 0.10 a | 23.9 ± 0.01 a | 70.02 ± 0.82 a |
| | $W_{80}F_{30}$ | 21.1 ± 0.31 b | 24.6 ± 0.04 b | 21.3 ± 1.37 b | 71.4 ± 1.49 a |
| 2016 | $W_0F_{10}$ | 21.5 ± 0.79 c | 12.2 ± 0.09 c | 12.2 ± 0.04 c | 45.87 ± 0.92 c |
| | $W_0F_{30}$ | 21.0 ± 0.61 c | 10.0 ± 0.07 d | 9.0 ± 0.04 d | 40.106 ± 0.83 d |
| | $W_{80}F_{10}$ | 26.5 ± 0.95 a | 16.9 ± 0.02 a | 16.2 ± 0.04 a | 59.54 ± 1.90 a |
| | $W_{80}F_{30}$ | 24.1 ± 0.82 b | 15.2 ± 0.06 b | 13.0 ± 0.04 b | 52.28 ± 0.716 b |

Note: pre-sowing irrigation ($W_{80}$) or no pre-sowing irrigation ($W_{80}$) and surface ($F_{10}$) or deep ($F_{30}$) fertilization. Data are the means of three replicates with standard errors and bars. Different letters indicate a significant difference at $p = 0.05$ according to Duncan's range test.

Simulation of biomass accumulation with respect to DAE was determined by formulas 2, 3, 4, 5, 6 and 7 (Table 2). In $W_{80}$ treatments, total biomass and above biomass fast accumulation period was prolonged by 2–7 d and 5–10 d, root and boll biomass accumulation at fastest accumulation period was shortened by 2 d and 4–5 d, respectively, compared with $W_0$ treatment. $W_{80}$ treatment had higher both total reproductive and vegetative organ biomass accumulation for maximum and average biomass accumulation rates during the fastest accumulation period than $W_0$ treatment. Under $W_{80}F_{10}$ total, stem, leaf and root biomass accumulation were extended by 1, 2, 10 and 1 d at fastest accumulation period compared with $F_{30}$. $W_{80}F_{10}$ had 13.9%, 12.5%, 10.9%, 15.0%, 17.5% 13.9%, 10.0% and 28.6%

higher maximum and average accumulation rates of total, aerial plant parts, boll, stem plus leaf and root biomass accumulation compared with $W_{80}F_{30}$ treatment.

### 3.5. Water-Nitrogen Productivity

Moisture consumption rate remained unaffected under both $W_{80}$ and $W_0$ treatment (Table 3). Compared with $W_0$, 8.1%, 31.1%, 52.6% and 39.2% greater root, stem plus leaf, bud plus boll and total biomass water productivity resulted in $W_{80}$ in all growth stages (Figure 5). $W_{80}F_{10}$ resulted in 32.0% and 15.2% higher total water and reproductive organs productivity respectively, compared with $F_{30}$ after 84 DAE.

Root nitrogen productivity had no significant difference under both $W_{80}$ treatment and $W_0$ treatment (Table 4, Figure 6). Nitrogen productivity of stem plus leaves, reproductive organs and total productivity increased by 31.3%, 42.9% and 23.1% in $W_{80}$ compared with $W_0$ at 54 to 99 DAE. $F_{10}$ produced 18.2%, 22.2% and 6.5% greater root, reproductive organs and total N productivity compared with $F_{30}$ from 54 DAE to 99 DAE.

### 3.6. Factors Affecting Productivity

Soil moisture content was positively related to nitrate reductase activity and available N, but had a negative relationship with root length, root dry matter, vegetative and reproductive organs dry matter accumulation (Figure 6). Water productivity of stem, leaf, bud and boll were negatively associated with soil moisture content and available N, but had a positive relationship with root dry matter, stem plus leaf dry matter and bud plus boll dry matter production. Root, stem and leaf water-N were positively related with bud plus boll water and N productivity.

Pathway analysis showed that root length, nitrate reductase activity had a strong direct effect on boll water-nitrogen productivity (Table 5). Nitrate reductase activity had higher indirect effect on bud plus boll water productivity through soil moisture content. Nitrate reductase activity had significantly indirect effect on bud plus boll nitrogen productivity through available nitrogen than root length. This shows that improved root distribution and physiological activities could directly, or indirectly enhance water-nitrogen productivity.

**Table 2.** Equation of cotton plant biomass accumulation under different irrigation and fertilization during 2015 and 2016.

| | Treatments | $R^2$ | $t_1$ (DAE) | $t_2$ (DAE) | T (DAE) | $t_m$ (DAE) | $V_m$ (g plant$^{-1}$ d$^{-1}$) | $V_t$ (g plant$^{-1}$ d$^{-1}$) | $W_0$ (g plant$^{-1}$) |
|---|---|---|---|---|---|---|---|---|---|
| | $W_0F_{10}$ | 0.975 | 32.32 | 62.11 | 29.79 | 46.62 | 0.97 | 0.85 | 45.47 |
| | $W_0F_{30}$ | 0.966 | 30.18 | 58.64 | 28.45 | 43.84 | 0.87 | 0.77 | 39.31 |
| Total Biomass of Cotton Plant | $W_{80}F_{10}$ | 0.99 | 34.20 | 65.46 | 31.26 | 49.21 | 1.23 | 1.08 | 60.54 |
| | $W_{80}F_{30}$ | 0.998 | 34.04 | 64.55 | 30.51 | 48.69 | 1.08 | 0.96 | 52.25 |
| | $W_0F_{10}$ | 0.986 | 35.92 | 75.49 | 39.57 | 54.91 | 0.42 | 0.37 | 26.4 |
| | $W_0F_{30}$ | 0.962 | 32.25 | 68.94 | 36.70 | 49.86 | 0.41 | 0.36 | 23.9 |
| Aerial Part Biomass of Cotton Plant | $W_{80}F_{10}$ | 0.987 | 42.34 | 86.00 | 43.66 | 63.30 | 0.51 | 0.46 | 35.50 |
| | $W_{80}F_{30}$ | 0.996 | 40.75 | 82.26 | 41.51 | 60.68 | 0.46 | 0.40 | 30.00 |
| | $W_0F_{10}$ | 0.954 | 30.19 | 73.85 | 43.66 | 51.14 | 0.35 | 0.31 | 24.09 |
| | $W_0F_{30}$ | 0.984 | 30.43 | 74.86 | 44.43 | 51.76 | 0.29 | 0.25 | 20.17 |
| Cotton Bud and Boll Biomass | $W_{80}F_{10}$ | 0.997 | 28.46 | 69.3 | 40.84 | 48.07 | 0.47 | 0.41 | 30.22 |
| | $W_{80}F_{30}$ | 0.997 | 28.84 | 70.35 | 41.51 | 48.76 | 0.40 | 0.36 | 26.33 |
| | $W_0F_{10}$ | 0.999 | 28.96 | 66.20 | 37.24 | 46.84 | 0.52 | 0.46 | 30.63 |
| | $W_0F_{30}$ | 0.988 | 34.17 | 70.87 | 36.70 | 51.79 | 0.36 | 0.32 | 20.70 |
| Cotton Stem and Leaf Biomass | $W_{80}F_{10}$ | 0.997 | 27.37 | 66.94 | 39.57 | 46.36 | 0.59 | 0.52 | 36.84 |
| | $W_{80}F_{30}$ | 0.996 | 25.85 | 54.96 | 29.11 | 39.83 | 0.59 | 0.52 | 26.90 |
| | $W_0F_{10}$ | 0.997 | 30.86 | 43.24 | 12.38 | 58.70 | 0.29 | 0.21 | 22.03 |
| | $W_0F_{30}$ | 0.999 | 29.58 | 38.96 | 9.38 | 61.30 | 0.28 | 0.17 | 20.80 |
| Cotton Root Biomass | $W_{80}F_{10}$ | 0.99 | 26.57 | 34.42 | 7.86 | 62.88 | 0.33 | 0.18 | 25.56 |
| | $W_{80}F_{30}$ | 0.999 | 23.97 | 30.69 | 6.72 | 63.92 | 0.30 | 0.14 | 23.81 |

Note: pre-sowing irrigation ($W_{80}$) or no pre-sowing irrigation ($W_{80}$) and base fertilization ($F_{10}$) or deep ($F_{30}$) application. DAE, indicates days after emergence (d). T1 and t2 are the beginning and termination days of the fast accumulation period, respectively. T indicates the duration of fast accumulation period. T = $t_2$ − $t_1$. $t_m$ is the after-emergence days of the maximum biomass accumulation speeds. Vm and Vt are the maximum and average biomass accumulation speeds during the fast accumulation period, respectively. $W_0$ is the maximum biomass accumulation.

**Table 3.** Change of total moisture consumption rate, total moisture consumption ratio, water productivity of root, water productivity of stem, leaf, water productivity of bud plus boll and water productivity of total dry matter under different irrigation and fertilization during 2015 and 2016.

| Year | Treatments | Total Moisture Consumption Rate ($cm^3$) | Total Moisture Consumption Ratio | Water Productivity of Root (g DM $cm^{-3}$) | Water Productivity of Stem and Leaf (g DM $cm^{-3}$) | Water Productivity of Bud and Boll (g DM $cm^{-3}$) | Water Productivity of Total Dry Matter (g DM $cm^{-3}$) |
|---|---|---|---|---|---|---|---|
| 2015 | $W_0F_{10}$ | 799.2 ± 14.62 b | 0.62 ± 0.01 b | 0.33 ± 0.02 a | 0.26 ± 0.00 c | 0.23 ± 0.01 b | 0.60 ± 0.03 c |
| | $W_0F_{30}$ | 867.5 ± 13.98 a | 0.69 ± 0.01 a | 0.29 ± 0.01 b | 0.19 ± 0.01 d | 0.15 ± 0.01 c | 0.42 ± 0.03 b |
| | $W_{80}F_{10}$ | 846.1 ± 16.39 a | 0.46 ± 0.01 d | 0.33 ± 0.01 a | 0.31 ± 0.02 a | 0.33 ± 0.01 a | 0.76 ± 0.03 a |
| | $W_{80}F_{30}$ | 859.1 ± 19.91 a | 0.55 ± 0.01 c | 0.34 ± 0.01 a | 0.28 ± 0.01 b | 0.25 ± 0.02 b | 0.66 ± 0.032 b |
| 2016 | $W_0F_{10}$ | 762.1 ± 13.42 b | 0.65 ± 0.01 b | 0.27 ± 0.01 bc | 0.15 ± 0.00 c | 0.15 ± 0.00 b | 0.57 ± 0.01 c |
| | $W_0F_{30}$ | 849.1 ± 28.84 a | 0.71 ± 0.02 a | 0.24 ± 0.00 c | 0.12 ± 0.00 d | 0.11 ± 0.00 c | 0.46 ± 0.02 d |
| | $W_{80}F_{10}$ | 822.7 ± 37.77 a | 0.54 ± 0.00 c | 0.31 ± 0.00 a | 0.20 ± 0.00 a | 0.19 ± 0.01 a | 0.69 ± 0.02 a |
| | $W_{80}F_{30}$ | 872.2 ± 36.93 a | 0.54 ± 0.02 c | 0.29 ± 0.00 ab | 0.18 ± 0.01 b | 0.16 ± 0.01 b | 0.62 ± 0.03 b |

Note: pre-sowing irrigation ($W_{80}$) or no pre-sowing irrigation ($W_{80}$) and surface ($F_{10}$) or deep ($F_{30}$) fertilization. Data are the means of three replicates with standard errors and bars. Different letters indicate a significant difference at p = 0.05 according to Duncan's range test.

**Table 4.** Changes in root nitrogen, stem plus leaf, nitrogen bud plus boll and total dry matter nitrogen productivity under different irrigation and fertilization during 2015 and 2016.

| Year | Treatments | Nitrogen Productivity of Root (g DM $mg^{-1}$) | Nitrogen Productivity of Stem and Leaf (g DM $mg^{-1}$) | Nitrogen Productivity of Bud and Boll (g DM $mg^{-1}$) | Nitrogen Productivity of Total Dry Matter (g DM $mg^{-1}$) |
|---|---|---|---|---|---|
| 2015 | $W_0F_{10}$ | 0.11 ± 0.004 b | 0.09 ± 0.000 b | 0.08 ± 0.003 c | 0.28 ± 0.007 c |
| | $W_0F_{30}$ | 0.11 ± 0.002 b | 0.07 ± 0.001 c | 0.06 ± 0.004 d | 0.24 ± 0.007 d |
| | $W_{80}F_{10}$ | 0.13 ± 0.000 a | 0.10 ± 0.005 a | 0.11 ± 0.001 a | 0.33 ± 0.005 a |
| | $W_{80}F_{30}$ | 0.11 ± 0.001 b | 0.11 ± 0.001 a | 0.09 ± 0.006 b | 0.31 ± 0.009 b |
| 2016 | $W_0F_{10}$ | 0.10 ± 0.003 b | 0.06 ± 0.000 c | 0.06 ± 0.000 c | 0.21 ± 0.004 c |
| | $W_0F_{30}$ | 0.09 ± 0.003 b | 0.05 ± 0.000 d | 0.04 ± 0.000 d | 0.18 ± 0.004 d |
| | $W_{80}F_{10}$ | 0.12 ± 0.00 a | 0.07 ± 0.000 a | 0.07 ± 0.000 a | 0.26 ± 0.008 a |
| | $W_{80}F_{30}$ | 0.10 ± 0.003 b | 0.07 ± 0.000 b | 0.07 ± 0.000 b | 0.23± 0.004 b |

Note: pre-sowing irrigation ($W_{80}$) or no pre-sowing irrigation ($W_{80}$) and surface ($F_{10}$) or deep ($F_{30}$) fertilization. Data are the means of three replicates with standard errors and bars. Different letters indicate a significant difference at *p* = 0.05 according to Duncan's range test.

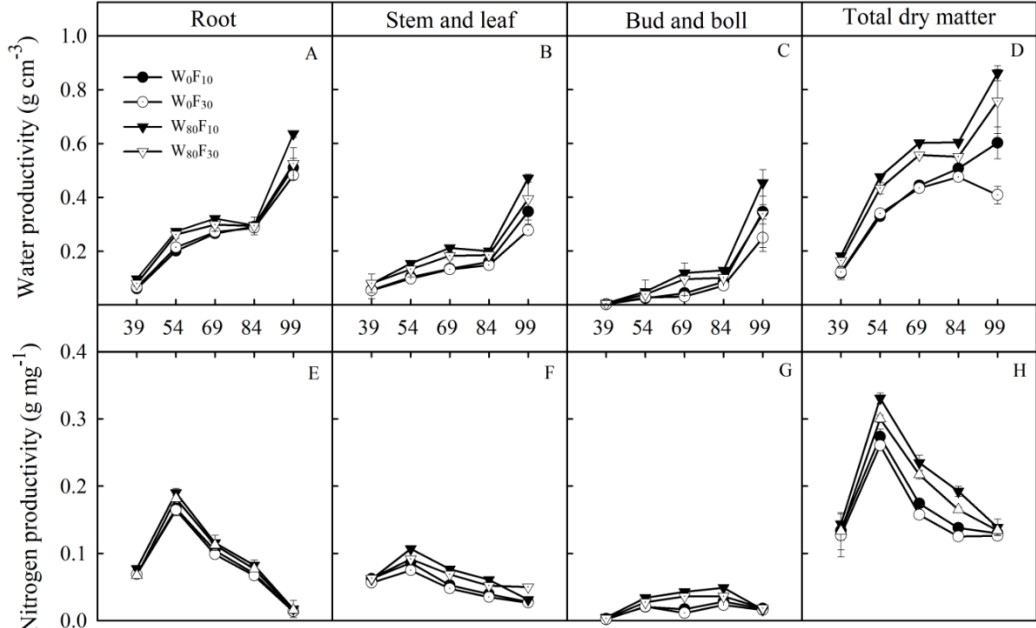

**Figure 5.** Dynamic change of root (**A**), stem and leaf (**B**), bud and boll (**C**) and total dry matter (**D**) water productivity and root (**E**), stem and leaf (**F**), bud and boll (**G**) and total dry matter (**H**) nitrogen productivity plant$^{-1}$ at pre-sowing irrigation ($W_{80}$) or no pre-sowing irrigation and basal surface fertilization ($F_{10}$) or deep ($F_{30}$) application with the days after emergence (DAE) in 2016. Bars indicate SD ($n = 3$).

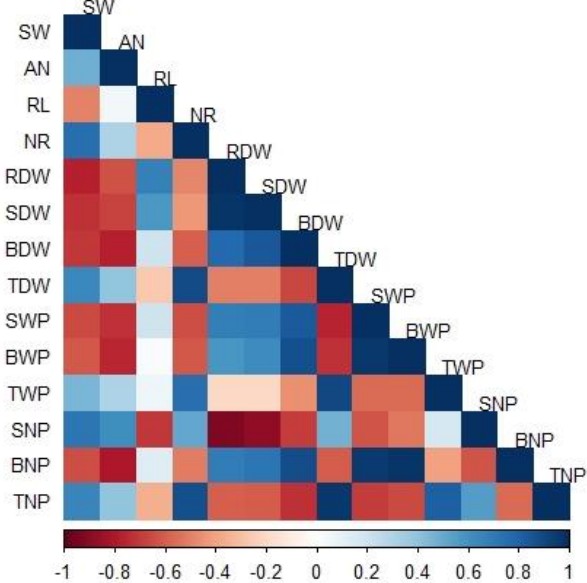

**Figure 6.** Correlation analysis ($n = 30$) of soil water content (SM), available nitrogen (AN, mg kg$^{-1}$), the root length (RL, cm), nitrate reductase (NR, μg g$^{-1}$FW h$^{-1}$), root dry matter (RDM, g plant$^{-1}$), stem and leaf dry matter (SDM, g plant$^{-1}$), bud and boll dry matter (BDM, g plant$^{-1}$) and total dry matter (TDM, g plant$^{-1}$) with nitrogen productivity and water productivity at pre-sowing irrigation ($W_{80}$) or no pre-sowing irrigation and base fertilizer surface ($F_{10}$) or deep ($F_{30}$) application with 2016. RWP, root water productivity (g DM cm$^{-3}$); SWP, stem and leaf water productivity (g DM cm$^{-3}$); BWP, bud and boll water productivity (g DM cm$^{-3}$); TWP, total water productivity (g DM cm$^{-3}$); RNP, root nitrogen productivity (g DM mg$^{-1}$); SNP, stem and leaf nitrogen productivity (g DM mg$^{-1}$); BNP, bud and boll nitrogen productivity (g DM mg$^{-1}$); TWP, total nitrogen productivity (g DM mg$^{-1}$).

**Table 5.** Path analysis (*n* = 60) for the direct or indirect effect on square and boll water-nitrogen productivity by soil moisture content, available nitrogen (mg kg$^{-1}$), the root length (cm), nitrate reductase (μg g$^{-1}$FW h$^{-1}$), root dry matter (g plant$^{-1}$), stem and leaf dry matter (g plant$^{-1}$) and bud and boll organ dry matter (g plant$^{-1}$) at pre-sowing irrigation or no pre-sowing irrigation and basal surface or deep fertilization in 2015 and 2016.

| | Square and Boll Water Productivity | | | | | | Square and Boll Nitrogen Productivity | | | | | |
| --- | --- | --- | --- | --- | --- | --- | --- | --- | --- | --- | --- | --- |
| | **x1-1** | **x2** | **x3** | **x4** | **x5** | **x6** | **x1-2** | **x2** | **x3** | **x4** | **x5** | **x6** |
| Soil moisture content (x1-1) | 0.058 | 0.41 | 0.455 | −0.183 | 0.943 | 0.121 | - | - | - | - | - | - |
| Available nitrogen (x1-2) | - | - | - | - | - | - | −0.027 | 0.211 | 0.375 | −0.947 | −1.221 | −0.062 |
| Root length (x2) | −0.177 | 0.382 | −0.128 | 0.735 | 1.548 | 0.031 | −0.013 | 0.312 | −0.117 | 0.616 | 0.638 | 0.274 |
| Nitrate reductase (x3) | 0.37 | 0.031 | 0.339 | 0.25 | 1.18 | −0.037 | −0.069 | −0.073 | 0.572 | 0.732 | 0.791 | −0.007 |
| Root dry matter (x4) | −0.277 | −0.056 | −0.165 | 0.564 | 1.633 | 0.226 | 0.143 | 0.13 | 0.179 | 0.201 | 1.765 | 0.055 |
| Stem and leaf dry matter (x5) | −0.259 | −0.047 | −0.149 | 0.512 | 2.688 | 0.033 | 0.154 | 0.111 | −0.151 | 0.471 | 1.801 | 0.152 |
| Bud and boll dry matter (x6) | −0.254 | −0.017 | −0.2 | 0.183 | 0.27 | 0.585 | 0.177 | 0.039 | −0.138 | 0.161 | 0.221 | 0.364 |

Note: "-" means no value.

## 4. Discussion

Water-nutrient application is an efficient strategy for improving plant performance under harsh environmental conditions i.e., drought stress, which ensures high cotton yield due to optimize root growth and activity in the soil [16,31]. In the present study, pre-sowing irrigation and basal surface fertilization significantly increased root distribution and physiological activity in the surface or deep soil profile at the boll setting stage. Improvement in these root traits contributed to greater shoot biomass and higher reproductive organ biomass accumulation led to greater water-nitrogen productivity.

Deeper root penetration can maximize soil moisture and nutrients uptake that can lead to maintain a high plant water and nutrient status [16,32,33]. We observed that pre-sowing irrigation and surface fertilization significantly increased root distribution and physiological activity in the surface soil (0–30 cm), indicating that improved the absorption and utility of water-nitrogen [34]. Because basal surface fertilizer application increased available water-N in the surface soil layer, which promoted cotton root distribution and physiological activity in the surface soil. This improved the absorption and utility of water-N and reduced the residual water-N in the surface soil profile. Moreover, root nitrate reductase activity in the deep soil profile (60–120 cm) enhanced, which indicated that decreasing root distribution regardless of improved root physiological activity in deep soil profile [20]. It is suggested that higher root distribution and physiological activity in both surface and deep soil profile could facilitate root and water-nutrient environment in the root zone, which can lead to higher root water-nitrogen absorption in cotton.

A strong relationship existed between root and shoot; shoots supply sufficient carbohydrates to roots that can develop and maintain root functioning which in turn can improve shoot growth by supplying a sufficient amount of nutrients, water and phytohormone. This further ensures crop productivity [5,35,36]. In this study, we observed that greater dry matter accumulated and allocation to the aerial parts has led to lower dry matter production in root and its physiological activity later in the season. The reason might be due to functional period of root (within 54–84 DAE) and the root biomass fast accumulation period (28–40 DAE) under different water-nitrogen management did not correspond. Root proliferation and physiological activity are positively associated with the root zone environment [32,33]. Therefore, pre-sowing irrigation and basal surface fertilization coordinates the relationship between root and water-nutrient in the soil. This in turn increased root absorptive capacity of water-nitrogen.

It is noteworthy that optimal water-nitrogen application could change the distance between water-nitrogen and root in the soil [16,37] as well as root physiological activity [33] to enhance the absorption of water-nitrogen. However, our data across the two years demonstrated that the water-nitrogen is an important management practice that can adjust the water-nitrogen productive ability in different plant organs which could result in greater water-nitrogen uptake. The possible reason might be improved root distribution and physiological activity in the surface soil and root physiological activity in the deep soil profile at 54–84 DAE promoted absorption of water-nitrogen from irrigation and deep layer water. This resulted in higher water-nitrogen productivity of reproductive organs at the boll setting stage. Secondly, root distribution and physiological activity could ensure the application of water-nitrogen, which increased leaf photosynthetic efficiency and leaf gas change parameters led to greater dry matter accumulation [22]. An adequate water-nitrogen in soil may decrease root distribution [15,20] and root dry matter at the fast accumulation period before 40 DAE. These phenomenon in turn decreased root dry matter accumulation and increased dry matter accumulation above ground parts at the boll setting stage (within 69–84 DAE).

Water use efficiency in terms of physiology is defined as the ratio transpiration and photosynthesis [38]. Lower dry mass accumulation in the aerial part can lead to a higher water-nitrogen use efficiency, but reducing water-nitrogen productive ability [10,32]. Interestingly, we observed that increasing root distribution and physiological activity in the surface soil layer and root physiological activity in deep soil layer at the boll setting stage can directly or indirectly promote dry mass accumulation and water-nitrogen productivity of the reproductive organs. More root distribution

can boost water use efficiency and drought resistance and consequently greater crop yield [33,39]. Higher root distribution led to a lower biomass accumulation in the aerial parts of crop plants [10,32]. We speculated that increasing root distribution and physiological activity can drive reproductive organs dry mass accumulation which results in higher water-nitrogen productivity of cotton crop.

## 5. Conclusions

Pre-sowing irrigation and surface basal fertilization could significantly promote reproductive organ biomass accumulation and productive ability of water-nitrogen. Pre-sowing irrigation combined with basal surface fertilization favored root morphological and physiological performance i.e., greater root biomass, longer root length in the surface soil profile (0–30 cm), higher root nitrate reductase activity in the surface or deep soil profile (60–80 cm) at the boll setting stage. Improvements in these root traits led to a higher water-nitrogen consumption, accumulation and allocation of reproductive structures of cotton plant. This in turn contributed to a higher water-nitrogen productive ability of the reproductive organ at the boll setting stage. These data highlighted that pre-sowing irrigation combined with basal surface fertilization is a promising option in terms of higher root morphological and physiological activity and water-nitrogen productivity of cotton crop in the arid region.

**Author Contributions:** Writing & conceptualization, Z.C.; data curation, H.G.; investigation, F.H.; writing—review & editing, A.K. and H.L.

**Funding:** This research was funded by the National Natural Science Foundation of China (31760355) and Program of Youth Science and Technology Innovation Leader of The Xinjiang Production and Construction Corps (2017CB005).

**Acknowledgments:** Shihezi University and the National Natural Science Foundation of China are acknowledged for their financial support to the study.

**Conflicts of Interest:** All the authors have approved the manuscript and agree with submission to your esteemed journal. There are no conflicts of interest to declare.

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
