# Peer review of "Pre-Sowing Irrigation Plus Surface Fertilization Improves Morpho-Physiological Traits and Sustaining Water-Nitrogen Productivity of Cotton"

_agronomy, doi:10.3390/agronomy9110772_

Round 1

Reviewer 1 Report

Line 2-5, please shorten the title

Line 35, correct to 73%

Line 78, correct 7.6 pH

Line 200-201, correct A 10% and 2%

Line 240, correct (Fig. 5).

Line 360 correct (Triticum aestivum L.)

Line Correct (Gossypium hirsutum L.)

Author Response

Dear Editor,

We sincerely thank the reviewers for their comments regarding our manuscript. We have carefully revised the manuscript according to their suggestions. In our point-by-point response below, the reviewers’ comments are in Black and our responses are in BLUE. The modifications in the revised manuscript are in RED. We hope these modifications are acceptable to you. Thank you again for your time and consideration.

Reviewer. 1

Line 2-5, please shorten the title

Response: Thank you very much for your comments. We have shortened accordingly in the revised version.

Line 35, correct to 73%

Response: Thank you very much for your comments. We have modified that on 35 lines and the other section in the text.

Line 78, correct 7.6 pH

Response: Thank you very much for your comments. We have modified that on 79 lines.

Line 200-201, correct A 10% and 2%

Response: Thank you very much for your comments. We have modified that on 203 lines and the other section in the text.

Line 240, correct (Fig. 5).

Response: Thank you very much for your comments. We have modified that in the text.

Line 360 correct (Triticum aestivum L.)

Response: Thank you very much for your comments.We have modified that in the reference list.

Line Correct (Gossypium hirsutum L.)

Response: Thank you very much for your comments. We have modified that in the reference list.

Reviewer 2 Report

Prior irrigation plus surface fertilization plays a key role in sustaining water-nitrogen productivity, reproductive organ biomass yield and root morpho-physiological traits in different soil profile of cotton

The title is really long and descriptive. First, prior irrigation does not represent its actual meaning i.e. pre-sowing irrigation. Secondly, ‘plays” should be “play” because of the plural subjects (irrigation plus fertilization). Lastly, if somehow you can group all the three impacts, that’ll be great. Also, “in different soil profile” can be totally get rid of.

Line 58: growing province in China

67% to the total

Do these two following lines concur with each other? Please clarify if there are inconsistencies.

Poor irrigation practices can develop a large root system and induce aging signals (such as, ABA), that can lead to low dry matter accumulation and water-nutrient productivity.

Post-sowing irrigation and snow melt can enrich deep-water layer (important soil moisture storage) in the soil. This can lead to a deeper root growth, enhance water uptake, improve photosynthetic capacity and reduces irrigation frequency

The standard procedure for presenting any water use data is recording all the relevant meteorological parameters at the study site. When we talk about water use/ET, it is not sufficient to provide only temperature and precipitation.

Line 109: Watermark sensors provide soil matric potential and not volumetric soil moisture. Can you detail the procedure (something like a soil water characteristic function) for conversion from former to latter?

And how did you convert that soil moisture (volumetric basis) to soil moisture (gravimetric basis)? Ideally you would need a bulk density measurement.

A lot of your procedures (even the basic ones) refer to other papers? I suggest you briefly describe them here.

Author Response

Prior irrigation plus surface fertilization plays a key role in sustaining water-nitrogen productivity, reproductive organ biomass yield and root morpho-physiological traits in different soil profile of cotton

The title is really long and descriptive. First, prior irrigation does not represent its actual meaning i.e. pre-sowing irrigation. Secondly, ‘plays” should be “play” because of the plural subjects (irrigation plus fertilization). Lastly, if somehow you can group all the three impacts, that’ll be great. Also, “in different soil profile” can be totally get rid of.

Response: Thank you very much for your comments. We have done as suggested.

Line 58: growing province in China

Response: Thank you very much for your comments. We have modified on 58 lines.

67% to the total

Response: Thank you very much for your comments. We have modified on 58 lines.

Do these two following lines concur with each other? Please clarify if there are inconsistencies.

Poor irrigation practices can develop a large root system and induce aging signals (such as, ABA), that can lead to low dry matter accumulation and water-nutrient productivity.

Post-sowing irrigation and snow melt can enrich deep-water layer (important soil moisture storage) in the soil. This can lead to a deeper root growth, enhance water uptake, improve photosynthetic capacity and reduces irrigation frequency

Response: Thank you very much for your comments. Poor irrigation practices, such as less irrigation rate, not proper time, and some irrigation in surface soil layer (0-30 cm), which easily lead to deep root growth and formation a large root system to enhance water uptake.  This can lead to a larger distribution of assimilate in root, and reducing the accumulation of assimilate in aerial parts or limiting the development of the shoot (Liu et al., 2008; Ball et al., 1994). Which consequently reduces yield.

Post-sowing irrigation after arable land promote irrigation water flowing to deep soil layer (about 60-80 cm soil layer), which reduced the evaporation of irrigation water, post-sowing irrigation helps root development because root development is mainly occur before flowering, which promotes the development of aerial parts and yield formation after flowering.

And the water storage in deep soil layer could directional induce root system distribution to deep soil layers (previous study), and avoiding a larger root system before flowering, relatively reducing the ratio of root to total dry matter, raising the accumulation of assimilate in aerial parts and yield formation.

Ball, R. A.; Oosterhuis, D. M., Mauromoustakos A. Growth dynamics of the cotton plant during water-deficit stress. Agron. J. 1994, 86,788-795.

Liu, R. X.; Zhou, Z. G.; Guo, W. Q., Effects of N fertilization on root development and activity of water-stressed cotton (Gossypium hirsutum L.) plants. Agric. Water Manag. 2008, 95, 1261-1270.

The standard procedure for presenting any water use data is recording all the relevant meteorological parameters at the study site. When we talk about water use/ET, it is not sufficient to provide only temperature and precipitation.

Response: Thank you very much for your comments. We have added the information about daily total solar radiation (MJ m-2), total precipitation (mm), maximum and minimum air temperature (°C) in the Figure 1. And we have changed the Table 1 to Figure 1 in the revised version.

Line 109: Watermark sensors provide soil matric potential and not volumetric soil moisture. Can you detail the procedure (something like a soil water characteristic function) for conversion from former to latter?

Response: Thank you very much for your comments. We only use the device (Time Domain Reflectometry (TDR) to measure soil water content for only irrigation, sorry to that. We have modified that on 109 lines.

And how did you convert that soil moisture (volumetric basis) to soil moisture (gravimetric basis)? Ideally you would need a bulk density measurement. A lot of your procedures (even the basic ones) refer to other papers? I suggest you briefly describe them here.

Response: Thank you very much for your comments. Soil moisture (volumetric basis) only was used for irrigation, because installed auxiliary equipment (a transparent polyethylene pipes, diameter was 5 cm) of TDR in the soil layer easily affected the root growth, so we only set 6 repetitions (only one auxiliary equipment in each plot) in each treatment to maintain the irrigation. Irrigation method and rate according to our previous study (Luo et al., 2014).

In addition, soil moisture (gravimetric basis) be used to analysis the relationship between soil water content, available nitrogen, root growth and nitrate reductase activity in different soil layer and other parameters. The irrigation each after four days according to the soil moisture (gravimetric basis) content to irrigation, which was determined by stoving method, negatively influenced root growth and the study.

Luo, H.; Zhang, H.; Han, H.; Hu, Y.; Zhang, Y.; Zhang, W., Effects Of Water Storage In Deeper Soil Layers on Growth, Yield, And Water Productivity Of Cotton (Gossypium Hirsutum L.) In Arid Areas Of Northwestern China. Irrig. Drain 2014, 63, 59-70.

Round 2

Reviewer 2 Report

All my concerns were taken care of or responded with justification.